# Optimizing the Phenylalanine Cut-Off Value in a Newborn Screening Program

**DOI:** 10.3390/genes13030517

**Published:** 2022-03-15

**Authors:** Dasa Perko, Barbka Repic Lampret, Ziga Iztok Remec, Mojca Zerjav Tansek, Ana Drole Torkar, Blaz Krhin, Ajda Bicek, Adrijana Oblak, Tadej Battelino, Urh Groselj

**Affiliations:** 1Clinical Institute for Special Laboratory Diagnostics, University Children’s Hospital, UMC Ljubljana, 1000 Ljubljana, Slovenia; barbka.repic@kclj.si (B.R.L.); ziga.remec@kclj.si (Z.I.R.); 2Department of Endocrinology, Diabetes and Metabolic Diseases, University Children’s Hospital, UMC Ljubljana, 1000 Ljubljana, Slovenia; mojca.tansek@kclj.si (M.Z.T.); ana.droletorkar@kclj.si (A.D.T.); tadej.battelino@kclj.si (T.B.); 3Faculty of Medicine, University of Ljubljana, 1000 Ljubljana, Slovenia; 4Department of Nuclear Medicine, UMC Ljubljana, 1000 Ljubljana, Slovenia; blaz.krhin@kclj.si (B.K.); ajda.bicek@kclj.si (A.B.); adrijana.oblak@kclj.si (A.O.)

**Keywords:** phenylalanine, phenylketonuria, newborn screening, DBS, NBS, PKU, cut-off value, false positive, true positive

## Abstract

Phenylketonuria (PKU) was the first disorder for which newborn screening (NBS) was introduced in the early 1960s. Slovenia started the NBS program for PKU in 1979, and the fluorimetric method was implemented in 1992, with a phenylalanine (Phe) cut-off set at 120 mol/L. This value has been in use for almost thirty years and has never been revised. We aimed to analyze the DBS samples and review the data from a large nationwide cohort of newborns to optimize the cut-off values for HFA screening to minimize the number of false positives while maintaining the highest level of sensitivity by detecting all those who needed to be treated. In the first prospective part of the study, we analyzed samples of all newborns in Slovenia in 2019 and 2020, and in the second retrospective part, we reviewed data from all known patients with hyperphenylalaninemia (HFA) in Slovenia born from 2000 to 2018. We defined true screening-positive cases as those that required a low-Phe diet. The sensitivity, specificity and positive predictive values of the modeling elevation of the Phe cut-off value from 120 µmol/L to 200 µmol/L were assessed. The number of recalls at the cut-off of 120 µmol/L was 108 out of 37,784 samples at NBS (2019–2020). Six newborns were defined as true positives and 102 samples as false positives. If the cut-off value was adjusted to 160 µmol/L, only 12 samples exceeded it and all six true positive newborns would be detected. Among the 360,000 samples collected at the NBS between 2000 and 2018, 72 HFA patients in need of a low-Phe diet were found. All the diagnosed cases would have been detected if the cut-off was set to 160 µmol/L. We demonstrated in a large group of newborns (400,000 in 20 years) that using the fluorimetric approach, a cut-off value of 160 µmol/L, rather than 120 mol/L, is safe and that there were no missing true positive patients who required treatment. By increasing the cut-off, this method becomes more precise, resulting in a significantly reduced rate of false positives and thus being less burdensome on both families and the healthcare system.

## 1. Introduction

Phenylketonuria (PKU; OMIM 261600) is the most prevalent inborn error of the metabolism (IEM) caused by mutations in the phenylalanine hydroxylase (PAH, OMIM * 612349) gene [1] which catalyzes the hydroxylation of phenylalanine (Phe) to tyrosine (Tyr) using tetrahydrobiopterin (BH_4_) as a cofactor [2]. Phe and its secondary metabolites accumulate in the blood and brain as a result of a PAH deficiency [2,3,4]. The phenotypes can vary from a very mild increase in blood Phe values to a severe classic phenotype with pronounced hyperphenylalaninaemia (HFA) [4]. PKU causes irreversible nerve cell damage if left undiagnosed and untreated, resulting in severe mental retardation, repressed language function, poor attention and underdeveloped motor control skills [5,6].

The development of Robert Guthrie’s diagnostic test in the early 1960s allowed mass screening for elevated Phe values, enabling early diagnostics of PKU. With the early introduction of effective dietary treatment, mental retardation due to PKU became very rare [3,7]. In Slovenia, the newborn screening program (NBS) for PKU was established in 1979 [8]. For 13 years, screening was performed using the Guthrie test [8,9,10]; after 1992, PKU was screened for using a fluorimetric method for the quantification of phenylalanine (Phe) in dried blood spots (DBS). 

To establish the Phe cut-off value, a pilot study of the NBS for PKU in Slovenia was conducted in 1992, when the fluorimetric method was adopted. The research involved roughly 7000 NBS samples. The Phe cut-off was set at 120 µmol/L. Additionally, the manufacturer of the utilized kit (PerkinElmer) specified a cut-off value of 127 µmol/L. The cut-off was rounded down to 120 µmol/L for safety concerns. This Phe cut-off value has been in use for almost thirty years and has never been revised. Cut-offs vary significantly among countries [6]; Slovenia’s is relatively low in comparison to others [11,12], with a high rate of sample recall.

Determining the optimal cut-off value is a critical feature of method efficacy in routine NBS practice. The sensitivity of the method should be as high as possible, meaning that the number of false negative results should be as low as possible while none of the true positive patients is missed [13]. 

We aimed to analyze the DBS samples and review the data from a large nationwide single-centre origin cohort of newborns to optimize the cut-off values for HFA screening to minimize the number of false positives while maintaining the highest level of sensitivity by detecting all those who needed to be treated.

## 2. Materials and Methods

### 2.1. Subjects

In the first, prospective part of the study, we analyzed DBS samples (and followed established protocol—Figure 1) from 37,784 newborns in Slovenia in 2019 and 2020. In the second, retrospective part of the study, we reviewed data on all newborns with HFA born between 2000 and 2018. Around 360,000 newborns were born in Slovenia during that period. The NBS program is a mandatory nationwide population program; the IEM registry is also mandated by national legislation [14], and written informed permission was not required from the families to be included in the NBS program or IEM registry. This study was performed in the context of regular NBS quality control, which is also required by the applicable legislation. The study protocol regarding the genotype-phenotype analyses in HFA patients was previously approved by the Slovene Medical Ethics Committee. Written informed consent was obtained from all the participants or their parents before genetic analysis was performed [10].

### 2.2. Specimen Collection, Methods, and Screening Protocol

Blood samples for NBS were taken between 48 and 72 h after birth from the newborn’s heel or by venipuncture [15] and collected on filter paper (Whatman 903, LKB, Austria). The DBS were sent to the Laboratory for Medical Radiochemistry, Department of Nuclear Medicine (LMR), where Phe quantification using a fluorimetric method was performed. The DBS controls (low and high) were included in every analytical batch to monitor the accuracy and precision within the system. From 2006 to 2018, the LMR was part of the United Kingdom Newborn Screening External Quality Assessment Scheme (KNEQAS) to ensure the quality of the results.

In LMR, a fluorimetric method was used involving a Neonatal Phenylalanine kit (PerkinElmer Life and Analytical Sciences, Wallac Oy, Finland). The method is based on the enhancement of the fluorescence of a phenylalanine-ninhydrin product using the dipeptide L-leucyl-L-alanine [16]. The reaction product was measured using a Fluorimeter 1420 VICTOR™ D Series (Perkin Elmer, MA, USA). A diagnostic algorithm following abnormal newborn screening results for PKU is presented in Figure 1.

### 2.3. Confirmation Analyses and PKU Classification

#### 2.3.1. Fluorimetric Method—S-Phe

All patients with Phe values above 120 µmol/L measured twice (i.e., the first and second DBS) or with Phe values greater than 200 µmol/L in the initial DBS were classified as NBS-positive and required further testing. Confirmatory analyses (Figure 1) were performed on blood serum in the Clinical Institute for Special Laboratory Diagnostics (CISLD), University Children’s Hospital Ljubljana (UCHL). The confirmation method is a fluorimetric in-house method, similar to the one in the LMR, based on the reaction between Phe (S-Phe) serum and ninhydrin in the presence of the dipeptide leucine-alanine. The reaction product is measured with a Perkin Elmer fluorescence spectrometer LS-55 (PerkinElmer, MA, USA). The reference interval for this method is between 50 µmol/L and 150 µmol/L. Patients with values above 200 µmol/L were also subjected to genetic testing; the PKU classification was based on their Allelic phenotype values (APV). Patients with values below 200 µmol/L were classified based on S-Phe values [3].

#### 2.3.2. Genetic Confirmation

Genetic analysis of the *PAH* gene was performed as previously described [2]. The classification of PKU for these patients was made based on Allelic phenotype values (APV), a model for genotype-based phenotype prediction in phenylketonuria, where a diagnosis is based on an allele with a lower APV value [17,18]. *PAH* variants determine the residual enzyme activity, which is the main determinant of the clinical phenotype in PKU. The less severe of the two *PAH* pathogenic variants—the variant with more residual enzyme activity and, therefore, with a higher APV value—usually determines the clinical phenotype of PKU [17]. 

#### 2.3.3. True Positive/False Positive

For the purpose of our study, all NBS-positive patients with Phe levels in areas where no dietary changes were needed were defined as false positives. True positives were all patients who required a low Phe-diet (Figure 1). 

All the patients were followed at the Department of Endocrinology, Diabetes, and Metabolic Diseases, UCHL.

### 2.4. Adjustment of the Cut-Off Value

Based on the data obtained from the phenotypic and genotypic analysis of 37,784 samples, we modeled increasing the lower Phe cut-off value in NBS to different values to determine the most appropriate in terms of greatest sensitivity, specificity, and positive predictive value. After adjusting the Phe cut-off, we re-evaluated the number of false positives. 

Additionally, we evaluated data from all individuals diagnosed with HFA between 2000 and 2018 who were also on a low-Phe diet. We examined how increasing the cut-off value of Phe to the value set in the first part of the study affects the number of false negatives in the second part of the study. 

## 3. Results

### 3.1. The First Part of the Study

We analyzed 37,784 DBS samples between 2019 and 2020. The number of recalls at the predetermined cut-off of 120 µmol/L was 108. Nine of them had a Phe concentration more than 200 mol/L, indicating that they were directly subjected to confirmatory analyses. The second DBS result exceeded 120 mol/L in seven out of 99 NBS borderline samples. As a result, 16 patients required additional confirmatory diagnostics. Genetic analysis of the *PAH* region was carried out in 11 patients, which included all those with a s-Phe value greater than 200 mol/L. Table 1 shows the patient’s number, Phe values, genotype, APV values, PKU classification based on the APV values or s-Phe, diet, and true/false positive determination.

#### 3.1.1. Detailed Description of 16 Patients

Patients 1 to 5 had elevated values of Phe in the DBS and they all had increased values of S-Phe. All had undergone genetic testing, and three of them had two different variants in the heterozygous state, all linked to PKU. Patients 4 and 5, who were twins, had variants linked to PKU in a homozygous state. Based on the APV values and the phenotype association of the PAH Variations [17,18], all five patients were diagnosed with classic phenylketonuria (cPKU), and they are all on dietary therapy based on a restricted Phe intake; thus, they are all defined as true positives.

Phe levels in patient 6’s DBS and blood serum were both elevated. Genetic testing revealed one PKU-related variant and one MHP-related variant. The patient has a brother with the same genotype and they are both on a low-Phe diet. The patient was diagnosed with mild phenylketonuria (mPKU) based on the APV values and was defined as a true positive.

Patient 7 had elevated Phe levels in both his DBS and blood serum. One pathological variant and one variant that maintains adequate enzymatic activity were discovered through genetic analysis. For 16 months, the patient was on a Phe-restricted diet. After that, he switched to a normal diet. Based on the APV values, the patient was diagnosed with mild hyperphenylalaninemia (MHP). The patient’s highest value was 310 µmol/L, indicating that the diet was likely started for safety concerns and some other clinical considerations. Later on, the levels fell below 120 µmol/L, necessitating the diet’s discontinuation. Due to the possibility that his diet was administered too early, we defined this patient as a false positive. 

Based on the APV values, the patient was diagnosed with mild hyperphenylalaninemia (MHP). The low-Phe diet may have been implemented too early for this patient and because he ceased dieting after 16 months, we defined him as a false positive. 

Patients 8 and 9 had elevated Phe values in their DBS and serum. Genetic analysis revealed one variant linked to PKU and one variant linked to MHP in both of them. The *PAH* genotype is the main determinant of the phenotype and the disease severity is in most cases determined by the milder of the *PAH* variants [17]. As a result, both were diagnosed with MHP. Since neither of these patients required a low-Phe diet, they were defined as false positives.

Patient 10 had borderline NBS results and increased S-Phe. The S-Phe values had dropped to normal in subsequent analyses, so there was no need for a low-Phe diet. Genetic analyses showed two variants in a heterozygous state, one linked to MHP and one with no APV value. The patient was defined as a false positive.

Patient 11 had borderline results using the fluorimetric method. The S-Phe values were slightly elevated; however, there was no need for further monitoring. 

Patients 12 to 15 had borderline results. The S-Phe values were within the normal range. They were all defined as false positives.

Patient 16 had a highly increased value of Phe. He also had pronounced hyperammonemia and hyperbilirubinemia. The baby died shortly after birth due to multiorgan failure. He had genetically confirmed gestational alloimmune liver disease (GALD). Although the fluorimetric method correctly measured the high Phe value, this patient did not have PKU; thus, we have not included him among the true positives.

The number of true positive patients was six, with the cut-off set to 120 µmol/L, so the number of false positives for the fluorimetric method was 102.

#### 3.1.2. Adjustment of the Cut-Off Value

We simulated the number of recalls with different Phe cut-off values ranging from 120 µmol/L to 200 µmol/L.

If we increased the lower cut-off value for PKU detection to 150 µmol/L, the number of recalls would be 16 instead of 108, and the number of false positives would be only ten instead of 102 out of 37,784 newborns. All six true positives would have been detected.

If we increased the cut-off value to 160 µmol/L, the number of recalls would be 12, with no false negatives. If we additionally increased the cut-off value to 180 µmol/L, the number of recalls would be 11. With the cut-off set at 180 µmol/L, we would miss patients number 10 and 15. However, they were both on a diet without restrictions. All six true positives would have been detected. Similarly, with the cut-off set to 200 µmol/L, the number of recalls would drop to nine, with the cut-off detecting all six true positives. For all the cut-off values, the number of false negatives is zero. All the results are presented in Figure 2.

The recall rate in the analyzed cohort was 0.29%. Table 2 shows the specificity, sensitivity, positive and negative predictive value, and detection rate at different Phe cut-offs.

### 3.2. The Second Part of the Study

We reviewed data from all patients who had been diagnosed with cPKU, mMPKU, or MHP, based on S-Phe values between 2000 and 2018. Among the 360,000 samples collected through the NBS, 72 HFA patients in need of a low-Phe diet were found. We had data on the Phe levels from the first DBS for all but two patients. We evaluated the number of false negatives with different Phe cut-off values, from 120 µmol/L to 200 µmol/L.

If we increased the cut-off value for PKU detection from 120 µmol/L to 160 µmol/L, all of those 70 patients would have been detected except for one, which means that the Phe value in his DBS was below 160 µmol/L. We have reviewed the medical documentation for that patient. The baby was born prematurely at 31 weeks of gestation and was in intensive care for 8 days, where he received two blood transfusions. The first DBS sample was taken after transfer to the neonatal department, following eight days of parenteral nutrition with an unknown composition and after two transfusions, which makes the Phe values questionable. According to the established protocol for gestational age and for being in intensive care, a newly collected sample must be taken after an additional 4 weeks. The new sample was taken after a few weeks and the Phe value was 390 µmol/L, thus this value is the correct one. In the further measurement of the S-Phe, the Phe values were above 600 µmol/L a few times, despite the introduced diet. Additionally, genetic analysis revealed two variants, NM_000277.1:c[1169A>G(;)1222C>T] with APV values of 6.8 and 0 (mPKU). This means that a false negative result for the patient would not be due to a cut-off value that was too high, but to premature sampling according to his gestational age; this patient was thus not considered a false negative.

If we additionally increased the cut-off value to 170 µmol/L and 180 µmol/L, we would miss one and three patients, respectively. All three patients have MHP. However, they all had S-Phe values of around 400 µmol/L or more; thus, this cut-off value might not be safe for PKU detection. At cut-off set at 200 µmol/L, we would miss six patients (Figure 3).

We ensured the measurement’s quality by participating in the external quality scheme UKNEQAS from 2006 to 2018. Evaluating the C-scores, the Phe values were stable, without significant fluctuations, with an average positive bias of 11.3% and a 95% confidence interval (CI) ranging from 10.6% to 12.0%. We took the 12% bias into account to ensure the safety of the newly set cut-off at 160 µmol/L.

## 4. Discussion

NBS is a system that identifies apparently healthy newborns with inherited disorders, usually metabolic in origin, before they cause serious morbidity or even death [9,19]. It was first introduced in the early 1960s in the USA with the Guthrie test for detecting PKU [20].

Different methods now exist for NBS for PKU worldwide, from enzymatic and bacterial inhibition assays to fluorimetric methods and methods using tandem mass spectrometry (MS/MS) [21]. Slovenia started the NBS program for PKU in 1979 and used the fluorimetric method since 1992, which is still in use in some Southeastern European countries [12]. Many NBS globally have Phe cut-off levels set higher than 120 µmol/L—e.g., 38.5% of 65 Latin American laboratories use 180 µmol/L or higher (up to 240 µmol/L) [11]. This is also apparent in Southeastern Europe, where most of the countries have a higher cut-off [22]. Additionally, cut-offs tend to vary significantly between countries, with no insight as to why such large differences exist [6].

In Slovenia, the cut-off value of 120 µmol/L has been used since 1992 and has never been revised. Thus, we analyzed DBS samples and reviewed NBS data for a wider group of neonates in order to optimize the cut-off values for PKU screening using the fluorimetric approach in terms of decreasing false positives while detecting all patients who required treatment. We included a prospective cohort of 37,784 neonates screened and a retrospective cohort of a considerably larger size, which included 70 patients who required nutritional treatment in the previous 18 years (identified among 360,000 newborns screened in that period). Both cohorts originate from a single center; the center covers both the NBS and HPA cohorts across the country, using the identical baseline circumstances and decision-making framework.

The central finding in the first part of the study was that at cut-off 120 µmol/L, the number of recalls and false positive samples is high. One hundred and eight samples were detected as borderline or positive while the number of true positive patients was only six.

The diagnosis and classification of PKU are usually done based on a Phe value in the blood before the patient starts a diet. This type of classification is not always straightforward in neonates, as the Phe values have not yet reached their highest values [3]. Due to this fact, in our study, the PKU classification was made based on the APV when possible.

For the purpose of our study, the definition of true or false positive was based solely on whether the patient required a low-Phe diet. The number of true positive patients was six using the fluorimetric method with a cut-off at 120 µmol/L and yielded 102 false positive samples. We consider it a false positive result when an additional sample is required that is later found to be normal, or positive-screening patients with Phe levels in areas where no dietary changes are needed.

False positive results disrupt the first weeks of a newborn’s life and can cause a great deal of anxiety and lasting emotional stress for the parents [23,24,25,26]. From the point of view of the physician and the laboratory analyst, the follow-up for each such result requires contacting the clinic for additional samples, testing them (which requires additional cost), and then notifying the clinic of the second result [13,25,27].

When we hypothetically increased the cut-off value for PKU detection from 120 µmol/L to 160 µmol/L, the number of recalls dropped from 108 to 12, with no false negatives. For Slovenia, this corresponds to approximately 50 recalls per 19,000 newborn children per year. Furthermore, cut-off values between 170 µmol/L and 200 µmol/L also proved to be safe and sensitive enough when applied to 37,784 newborns.

To assess the suitability of the various cut-off values determined in the first part of this study, we applied them to 70 HFA patients, who required a low-Phe diet, diagnosed among 360,000 newborns between 2000 and 2018. When we raised the cut-off to 160 µmol/L, all of those patients would have been detected except for one, who was later found to have had a blood sample taken too soon for his gestational age and was also on two transfusions and parenteral nutrition. A variety of outside factors could also have influenced the Phe values in this newborn’s blood. The first is that he was born at 31 weeks of gestation, implying that the Phe values could have been falsely increased or decreased. The second reason is that he was on a parenteral diet, which produces false positive results by increasing the blood Phe levels. The third component is transfusion, which is not supposed to affect the Phe values; however, when all three factors are combined, the Phe values, as well as other analytes, can become questionable [15,28,29,30,31]. If we additionally increased the cut-off values to 170 µmol/L and 180 µmol/L, we would miss one and three patients, respectively, all of whom had S-Phe values around 400 µmol/L. Currently, all three patients are on a non-strict diet, only avoiding high-protein foods in large quantities. Evidence of the need to start treatment with blood Phe concentrations between 360 μmol/L and 600 μmol/L is currently inconsistent, but due to caution, there is a recommendation for the treatment of children under 12 years of age with Phe values above 360 µmol/L [32]. Thus, it is not yet entirely clear whether or not our false negatives would have any long-term consequences. Nevertheless, cut-off values of 170 µmol//L or 180 µmol/L will not be safe for PKU detection.

Despite that PKU screening is now almost universaly introduced accross the Europe, cut-off values for PKU detection vary widely between different countries [6,33]. The time of the blood sampling and the methods being used have changed over time, with more specific MS/MS technology being used increasingly, especially in developed countries [21]. Usually, laboratories have to determine their own cut-off values. The next step could be to compare them with international databases, like CLIR (Collaborative Laboratory Integrated Reports) [34]. However, most countries still have a fixed cut-off for Phe, which is also the sole biochemical biomarker to screen for *HPA*, which leads to reduced complexity compared to some other disorders relying on multiple biomarkers [32].

The new cut-off values were determined using 37,784 samples, six of which were true positives. We acknowledge that this number may be low, and it was only by integrating data from the second cohort, which had 70 true positives, that we were able to determine if the set values were acceptable. Thus, we believe that the study’s total cohort of 400,000 newborns and 76 true positives in the analyzed period quite enables us to draw valid conclusions in this study.

The lower than usual cut-off (120 µmol/L) in our program compared to most other NBS programs allowed us to test a wider spectrum of possible cut-offs compared to other programs. The strong variability of Phe cut-offs across countries suggests a lack of data and consensus on what is optimal and safe for the Phe cut-off.

Each NBS program adheres to an ethical standard of adjusting the cut-off to ensure that it not only detects all patients who may eventually require nutritional therapy, but also burdens the fewest possible other families. We already adjusted the Phe cut-off for our program based on our study, which was one of the study’s primary goals.

## 5. Conclusions

We demonstrated in a large group of newborns (400,000 in 20 years) that using the fluorimetric approach, a cut-off value of 160 mol/L, rather than 120 mol/L, is safe and that there were no missing true positive patients who required treatment. By increasing the Phe cut-off, this method becomes more precise, resulting in a significantly reduced rate of false positives and so being less burdensome on both families and the healthcare system.

## Figures and Tables

**Figure 1 genes-13-00517-f001:**
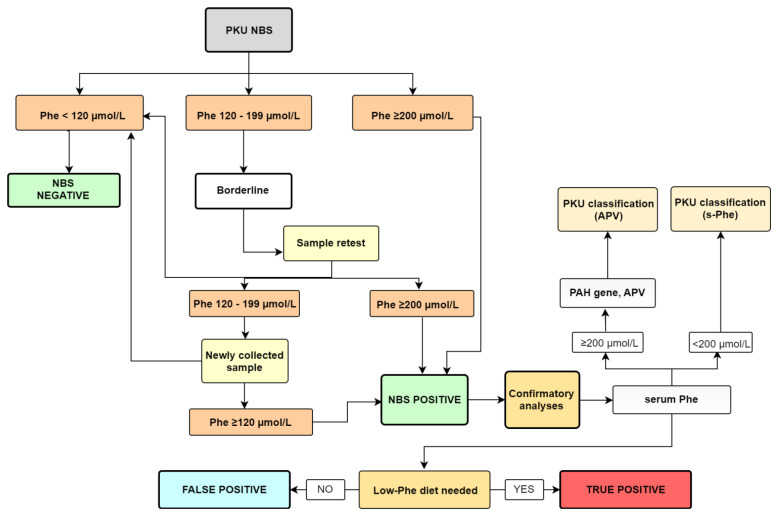
A diagnostic algorithm following abnormal newborn screening results for PKU.

**Figure 2 genes-13-00517-f002:**
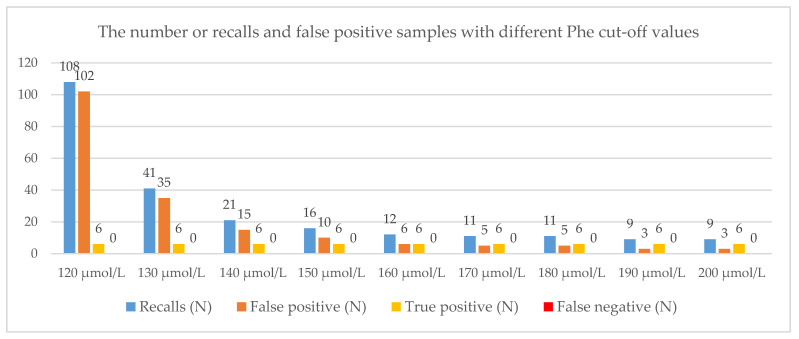
The number of recalls, false positives, true positives, and false negatives at different cut-off values; number of samples: 37.784, number of true positives: 6.

**Figure 3 genes-13-00517-f003:**
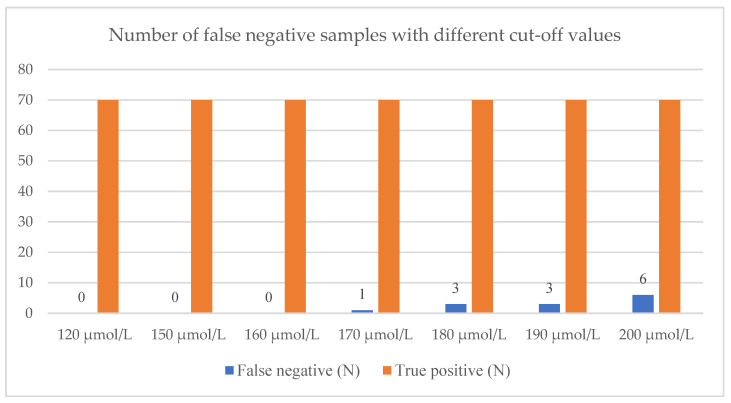
The number of false negatives with different cut-off values.

**Table 1 genes-13-00517-t001:** Phe values, genotype, and diagnosis of 16 newborns who needed PKU confirmatory analyses.

Patient	DBS-Phe (µmol/L)	DBS-Phe (µmol/L)— New Sample	S-Phe (µmol/L)	Genotype (*PAH* Gene)	APV Allele1/ Allele2	* PKU Classification	Diet	TP
1	500	nd	569	NM_000277.1:c[143T>C(;)913-7A>G]	2.1/0	cPKU	yes	yes
2	420	nd	1118	NM_000277.2:c[143T>C(;)1222C>T]	2.1/0	cPKU	yes	yes
3	550	nd	802	NM_000277.2:c[842C>T(;)1222C>T]	0.8/0	cPKU	yes	yes
4	490	nd	751	NM_000277.3:c[473G>A];[473G>A]	0/0	cPKU	yes	yes
5	580	nd	983	NM_000277.3:c[473G>A];[473G>A]	0/0	cPKU	yes	yes
6	290	nd	593	NM_000277.1:c[442-5C>G(;)842C>T]	6.2/0.8	mPKU	yes	yes
7	310	nd	241	NM_000277.2:c[58C>T(;)165T>G]	0/8.1	MHP	no	no
8	180	200	229	NM_000277.3:c[473G>A(;)827T>C]	0/10	MHP	no	no
9	260	nd	250	NM_000277.3:c[1208C>T(;)1222C>T]	9.3/0	MHP	no	no
10	150	120	247	NM_000277.3:c[678G>C(;)734T>C]	nd/9.9	MHP	no	no
11	150	170	194	/	/	MHP	no	no
12	130	180	131	/	/	MHP	no	no
13	130	120	92	/	/	/	no	no
14	130	130	130	/	/	MHP	no	no
15	180	160	144	/	/	MHP	no	no
16	710	nd	417	/	/	GALD	/	/

DBS-Phe—the phenylalanine (Phe) value in the dried blood spot (DBS); DBS-Phe—new sample—the Phe value in newly collected DBS; S-Phe—the Phe value in serum; APV—Allelic Phenotype Values; TP—true positive. * Based on APV cut-off values: 0–2.6 = cPKU; 2.7–6.6 = mPKU; 6.7–7.5 = mPKU-MHP; 8–10.0 = MHP [17]. nd—not determined. When the *PAH* gene was not analysed, PKU classification was based on s-Phe [4]. cPKU—classic phenylketonuria; mPKU—mild phenylketonuria; MHP—mild hyperphenylalaninemia; GALD—gestational alloimmune liver disease.

**Table 2 genes-13-00517-t002:** Specificity, sensitivity, positive and negative predictive value, and detection rate at different Phe cut-offs.

Number of newborns	37,784								
Phe cut-off value (µmol/L)	120	130	140	150	160	170	180	190	200
Recalls (N)	108	41	21	16	12	11	11	9	9
False positive (N)	102	35	15	10	6	5	5	3	3
True positive (N)	6	6	6	6	6	6	6	6	6
False negative (N)	0	0	0	0	0	0	0	0	0
True negative (N)	37,676	37,743	37,763	37,768	37,772	37,773	37,773	37,775	37,775
Detection rate (%)	0.019	0.019	0.019	0.019	0.019	0.019	0.019	0.019	0.019
Sensitivity (%)	100	100	100	100	100	100	100	100	100
Specificity (%)	99.73	99.91	99.96	99.97	99.98	99.99	99.99	99.99	99.99
Positive predictive value (%)	5.56	14.63	28.57	37.50	50.00	54.55	54.55	66.67	66.67
Negative predictive value (%)	100	100	100	100	100	100	100	100	100

## Data Availability

Not applicable.

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
