# Peer review of "Optimizing the Phenylalanine Cut-Off Value in a Newborn Screening Program"

_genes, 2022, doi:10.3390/genes13030517_

Round 1

Reviewer 1 Report

I read with interest your manuscript on optimizing the cut-off levels of phenylalanine in your newborn screening program decreasing the number of false-positives and increasing the specificity of your screening method.  Your study results show clearly the improvement of performance of the newborn screening for PKU in your population solely by increasing the cut-off level of phenylalanine.

Comment 1:  concerning the confirmatory analysis: why do you not use LC-MSMS for confirmatory analysis of Phe (and Tyr) in serum as you have the equipment available for the monitoring of patient's dietary status?  

Comment 2:  patients with Phe-levels between 120-600 micromol/L are defined as mild hyperphenylalaninemia are picked up by all newborn screening programs.  Why do you classify them under "false-positives" instead of MHP?  Even when these patients do not need dietary treatment in childhood and adolescence a Phe level of >360 micromol/L can have great impact in women who want to get pregnant.

Comment 3:  Why didn't you calculate the sensitivity, specificity and positive predictive value in addition to the absolute numbers of false-positive linked to the cut-off values of 120 and 160 micromol/L?  

Comment 4:  in the second part of the study you mentioned 1 false-negative PKU due to early blood sampling, blood transfusions and TPN.  Please rephrase the comments on this patient giving more details on the interfering circumstances and their relative consequences on the Phe level: I presume that blood transfusion(s) already give a high rate of false-negative result for all disorders included in a newborn screening program.

Some minor comments on typo's and english language.

Reviewer 2 Report

Dear  authors,

I have read your paper with interest.

Please find my critique below. 

This would all be resolvable, however, your paper has three major flaws.

Figure 1:

The figure indicates that a false positive screening result is defined by a positive screening result based on a) Phe-concentration, b) genetic confirmation by evaluating the PAH-gene and c) dependence on a Low Phe-diet. This makes the Low-Phe diet a part of the screening test which it is not. This culminates in the questionable statement of line 107, True positives were all patients, who required low Phe diet (Figure 1). This is true by definition, as false positives, by definition not patients, are defined as not needing a Low Phe diet. Table 1 also wrongfully suggests that the diet determines whether a patient is a true positive or not. This is a first major flaw of your paper.

2)

While the definition of true positive, true negative, false positive and false negative is already wrong, simulating different cut off values with so little positives and especially cases is another major flaw of this paper (page 7, top part). An evaluation of cut-off values can only be sensibly done if the authors are able, for every cut off, to present detection rate, false positive rate, positive predictive value and negative predictive value. This is actually a combination of two flaws

3)

To do the same on a population of subjects originating from analyses between 2000 and 2018 is a third major flaw. To do this, the you should show that through these years, the Phe measurements in your laboratory had been stable, and no major upward and downward trend observed. However, in neonatal laboratory practice this is almost always the case.

I am sorry to say that you, the authors are in my view on a wrong track here. Your study population is to small and your approach flawed in multiple ways.

One sensible way of comparing the cut off values is by comparing them to those from international peers, there is an enormous amount of literature, but there are also international databases available (CLIR-Mayo Clinics) and many international interlaboratory analytical proficiency sample schemes could be of use. None of this is even remotely touched upon, e.g. in the discussion.

Other major points of critique are

1)

The level of English is not up to standards-this may be easily resolved by having the manuscript tended to by a scientific writer or maybe a colleague who is a native speaker.

2)

Line 52-You base  this statemen on a reference from1998. I think it is fair to say that most screening for Phe is done with tandemMS. Happy to accept that this method is still frequently used but then the authors need to produce recent refs, 2015 or later.

3)

There is not a very good connection between the methods and result section. For instance, the Methods section mentions several confirmatory tests, (par. 2.3) but whether all of these or just a selection was used in the subjects of 3.1 first Alinea, is unclear.

4)

Par 3.1 is extremely hard to follow without a flow diagram indicating which subjects were positive in the screening tests and which confirmatory tests were subsequently done.

5)

Par 3.1 goes on the describe patients in various combinations, indicated by a number, and the reader needs to figure out that these numbered patients are summarized in table 1.

Round 2

Reviewer 1 Report

I read through the revised paper on Optimizing phenylalanine cut-off value in a newborn screening program.  The content looks good and is well improved compared to the first submission.  I suggest to work on the english language before the paper can be accepted to be published.

Author Response

we have sent the paper to a native English speaker for editing. 

Reviewer 2 Report

Dear authors.

I have read your revised manuscript and your rebuttal to my critique with interest and there remain issues with your handling my critique.

Response 1: you correctly identify programs globally that have higher cut-offs-I do not agree that these COVs vary much and that there is no proper funding for those COVs. If anything, your COV is rather low-and you should explain in your introduction what this COV is based on. You should also compare your referral rate with that of other large and long running published programs, using these as a bench mark. You will notice that your original referral rates are high.  

You have adapted the scheme in figure 1, that is now comprehensible.

You have now produced a table with performance indicators but this will not resolve the fact that your number of true positives in this cohort is too low.

You have satisfactorily answered to my question on long term QC to indicate stability of methodology.

You have not satisfactorily answered my point 4. I think you should produce reliable international benchmarks-I am certain that colleagues at Mayo Clinics could help you with that and you can search the international literature. You can draw valid conclusions on your population of 400.000 analyses but you actually based your cut off on a much smaller cohort of about 38000.

I am confused by your remark that a cut off of 120 ug/l allows to test a wider spectrum of possible cut-offs as compared to other programs, which miss part of the individuals with HPA with mild Phe elevations. That should never be a reason to determine a CO, it is at the brink of being unethical, and the mild cases are the ones that screening programs, including yours intend not to find, so why bother?

I feel the English is still unsatisfactory-adding an article here and there does not do the trick but I will leave this to the editors.

That brings me to the current, revised version of the document.

  • I do not understand why you have split your cohort in two parts. You should give a rationale for that. If there is no rationale, please simplify the paper by analysing the entire 2000-2018 cohort and the 38000 together. That would give enough cases to draw proper conclusions
  • Ln 121: the definition of positive is entirely based on requiring low Phe diet. How was this determined for individual cases? What was the diagnostic call?
  • While there is a table on the details of patients in the 38000, there is not in the 2000-2018 cohort. Were the positive patients exclusively the patients that needed a low Phe diet? This table, similar to table 1, must be added.
  • Ln 141-why are not all patients genotyped, especially as the APV values seem to determine true positivity?
  • Ln 164-according to the definition of the authors (2.3.3) patient 6 is not low-Phe diet-dependent and thus a false positive.
  • Table 2-Number of patients: these are not patients.
  • 3.2 line 25. How have these patients been defined? Were they all on Phe low diet, as per definition in 2.3.3? Were they genotyped? One would expect that the mPKU patients were not? In this respect, it is important to calculate the incidence of PKU in this cohort and compare to other, e.g. European cohorts. To make a proper comparison, the authors must scrutinize these patients in the same way they did in table 1.
  • Pg 9-Ln 63. Just an example-please remove article before NBS.
  • Ln 78. “Accordingly, most 76 of the NBS programs globally are missing a substantial part of the individual MHP, be-77 cause it is considered that it is clinically not important to detect them. The main goal of 78 most NBS programs is thus not to detect all of the individuals with MHP, but not to miss 79 those who eventually need dietary intervention.”

    This seems to be exactly what the described program is doing with the low cut off.
  • Ln 90. “Phe values have not yet reached their highest values [4].” But at a sampling time of 48-72 hrs they have.
  • Ln 91: “in our study, the PKU classification was made based on APV when possible”. This is in contradiction with 2.3.3.
  • Ln 94: “We consider a false positive result when an additional sample is required, which is later found to be normal or screening-positive patients with Phe levels in areas where no dietary changes are needed.

    This is very important as it defines positives and influences the entire study – what are the Phe levels where no dietary changes are needed?
  • Ln99-100: re testing after a high Phe result is not common in many programs. Most programs refer on the primary sample.
  • Ln 144-146. “The lower than the usual cut-144 off (120 μmol/L) in our program as compared to most”.
    You want to miss the cases with mild Phe elevations and being able to experiment with cut off values is not the reason the 120 was chosen and shouldn't be. The reason to do screening is not to do research, but to properly screen. The authors should indicate what the original cut off was based on. The point that they raise the cut off is valid, as it was low by international comparison. Their reasoning was off and still is.

I think a complete re-writing and simplification of this paper is eminent. A proper line could be

  • We historically applied a cut-off value of 120 ug/l blood (based on….)
  • We had indications that this was not OK (mention indications)
  • We tried higher cut off in our historic data (one cohort).
  • Based on the results together we came up with a new cut-off of…

Where you use a single definition of a positive case and describe all 72+16 = 88 patients in a table (or supplementary table).

Author Response

Reply to Reviewer 2- (round 2)

Dear authors. I have read your revised manuscript and your rebuttal to my critique with interest and there remain issues with your handling my critique.

You correctly identify programs globally that have higher cut-offs-I do not agree that these COVs vary much and that there is no proper funding for those COVs. If anything, your COV is rather low-and you should explain in your introduction what this COV is based on.

Reply: Cut-offs vary significantly among countries, as demonstrated by a 2020 research that examined studies including 119,152,905 individuals done between 1964 and 2017 (Shoraka, H.R.; et al. Global prevalence of classic phenylketonuria based on neonatal screening program data: Systematic review and meta-analysis. Korean J. Pediatr. 2020, 63, 34–43, doi:10.3345/kjp.2019.00465). We want to indicate in this phrase that there is no information as to why there are such huge variations between countries.

To establish the Phe cut-off value, a pilot study of the NBS for PKU in Slovenia was conducted in 1992, when the fluorimetric method was adopted. The research involved roughly 7,000 NBS samples. The Phe cut-off was set at 120 µmol/L. Additionally, the manufacturer of the utilized kit (PerkinElmer) specified a cut-off value of 127 µmol/L. The cut-off number was rounded down to 120 µmol/L for safety concerns. This value has been in use for almost thirty years and has never been revised. We have added this information in the Introduction (please see page 2, lines 61).

You should also compare your referral rate with that of other large and long running published programs, using these as a bench mark. You will notice that your original referral rates are high.

Reply: Unfortunately, we were unable to find any well-comparable data on referral rates in other countries, but we found data on HPA incidences and Phe cut-offs. We agree that our recall was high, reflecting the significant number of false positives, as well as the fact that a considerable proportion of these had DBS Phe values were 120 µmol/L, on or just above the cut-off.

You have now produced a table with performance indicators but this will not resolve the fact that your number of true positives in this cohort is too low.

Reply: The number of true positives in the prospective part of the study is a fact that was measured and it was noted among study limitations (please see page 10, lines 133). That is why we evaluated the acquired cut-offs on an additional 70 patients on a low-Phe diet in the retrospective part of the study, basing on an adequate number of children screened. Thus, we believe that the study's total cohort of 400,000 newborns and 76 true positives in the analysed period quite enables us to draw valid conclusions in this study (in the context of clear set of limitations stated). We can agree, that this cohort might not be ideal to solve this issue beyond any doubt. However, having in mind the general lack of data and recommendations on this particular issue, and wide international existing variations, we still believe that our study could be a relevant contribution to the field and might at least help informing and inducing further studies. We firmly believe this is an important, yet unsolved scientific question.

You have not satisfactorily answered my point 4. I think you should produce reliable international benchmarks-I am certain that colleagues at Mayo Clinics could help you with that and you can search the international literature.

You can draw valid conclusions on your population of 400.000 analyses but you actually based your cut off on a much smaller cohort of about 38000.

Reply: We prospectively included a cohort of 37,784 newborns screened, and a much larger retrospective cohort, which included additional 70 patients that required dietary therapy in the last 18 years (identified among 360,000 newborns screened in that period). Both cohorts are of single-centre origin; the centre is covering both nation-wide NBS and HPA cohorts, under same baseline conditions and with same decision-making framework. For both the prospective and retrospective cohorts the newborn screening Phe cut-off was the same (120 µmol/L), which enabled us to simulate raising the cut-off to test also the possible higher Phe cut-offs in both the cohorts. But we are unable to combine the both cohorts because the cohorts were not assembled in the same way and are independent and for the retrospective cohort not all the data would be available. Nevertheless, we believe this approach still allows us to draw sufficiently based conclusions.

I am confused by your remark that a cut off of 120 ug/l allows to test a wider spectrum of possible cut-offs as compared to other programs, which miss part of the individuals with HPA with mild Phe elevations. That should never be a reason to determine a CO, it is at the brink of being unethical, and the mild cases are the ones that screening programs, including yours intend not to find, so why bother?

Reply: We agree with your point; actually, this notion was among main reasons, why we conducted this study. We are ethically obliged to adjust the cut-off to the level so it not just detects all the patients eventually needing dietary therapy, but also that it burdens as few other families as possible. Based on our study we already adjusted Phe cut-off for our program, but we would also like to possibly help any other programs to possibly adjust their cut-off or at least to start thinking if their cut-off is really adequate. Based on the great spectrum of cut-offs globally, many of the programs probably have much room for improvement, because all of them cannot be right at the same time.

I feel the English is still unsatisfactory-adding an article here and there does not do the trick but I will leave this to the editors.

Reply: We have sent the manuscript for professional English editing (native speaker).

That brings me to the current, revised version of the document.

  • I do not understand why you have split your cohort in two parts. You should give a rationale for that. If there is no rationale, please simplify the paper by analysing the entire 2000-2018 cohort and the 38000 together. That would give enough cases to draw proper conclusions

Reply: As already argued above, the first section of the study is prospective, meaning that we conducted analyses and so have all of the data for these 37,000 babies, including Phe in DBS, S-Phe, and genetic data (for those with S-Phe over 200 µmol/L). The second part of the study is a retrospective cohort, which included additional 70 patients that required dietary therapy in the last 18 years (identified among 360,000 newborns screened in that period). For both the prospective and retrospective cohorts the newborn screening Phe cut-off was the same (120 µmol/L), which enabled us to simulate raising the cut-off to test also the possible higher Phe cut-offs in both the cohorts. But we are unable to combine the both cohorts because the cohorts were not assembled in the same way and are independent and for the retrospective cohort not all the data would be available. Nevertheless, we believe this approach still allows us to draw sufficiently based conclusions.

  • Ln 121: the definition of positive is entirely based on requiring low Phe diet. How was this determined for individual cases? What was the diagnostic call?

Reply: We follow applicable international guidelines. In principle, diets are initiated when s-Phe levels exceed 360 µmol/L. However, we also try to make individual decisions (depending on age of newborn, nutritional status, Phe values fluctuations and newborns overall health status etc.). Throughout the period of the study, these decisions were made by one of only three physicians (metabolic paediatricians), all of who are co-authors of this study. After additional review, we agree that patient number six stands out here, as you also indicated. The patient's highest value was 310 µmol/L, indicating that the diet was likely started for safety concerns and some other clinical considerations. Later on, the levels fell below 120 µmol/L, necessitating the diet's discontinuation. Due to the possibility that his diet was administered to early, we later defined this patient as a false positive. Thus, the new total of true positives in the first section of the research is six. Due to the greater transparency of Table 1, we have changed the numbering of patients 6 and 7. As a result, patient 6, who is now a false positive, has been reclassified as patient 7.

  • While there is a table on the details of patients in the 38000, there is not in the 2000-2018 cohort. Were the positive patients exclusively the patients that needed a low Phe diet? This table, similar to table 1, must be added.

Reply: Correct; per definition, true positive patients were only those who need a low-Phe diet. In our first cohort, the calculated recall rate was 0.29%. Thus, we might conclude that during the course of 20 years and 400,000 infants, about 1,150 neonates had Phe levels in DBS equal to or more than 120 µmol/L. Thus, a table (such as Table 1) would have data on over 1000 newborns who tested borderline or positive for PKU on the NBS. As previously stated, data for all of these newborns are either missing or incomplete, and thus gathering those data would generate an unnecessarily large amount of labor. As a result, we conducted a rigorous analysis on a smaller cohort with complete data and then confirmed the findings in an additional 70 HFA patients on a low-Phe diet. Further methodological concerns are also explained above.

  • Ln 141-why are not all patients genotyped, especially as the APV values seem to determine true positivity?

Reply: In our study, APV values are not used to determine true positives; rather, they serve as a classification tool for PKU (cPKU, mPKU, MHP). (cPKU, mPKU, MHP).

All NBS positive patients with Phe levels in areas where no dietary adjustments were required were considered as false positives for the purposes of our study. All patients who required a low Phe-diet were true positives. Only those patients with s-Phe values greater than 200 µmol/L were genotyped (established protocol for last three years).

Ln 164-according to the definition of the authors (2.3.3) patient 6 is not low-Phe diet-dependent and thus a false positive.

Reply: We concur with your remark. As previously stated, the low-phe diet may have been implemented to early for this patient (for safety concerns), and because he ceased dieting after 16 months, we classified him as false positive. Due to the greater transparency of Table 1, we have changed the numbering of patients 6 and 7. As a result, patient 6, who is now a false positive, has been reclassified as patient 7.

  • 3.2 (second part of a study) line 25. How have these patients been defined? Were they all on Phe low diet, as per definition in 2.3.3? Were they genotyped? One would expect that the mPKU patients were not?

Reply: This part of a study was retrospective, meaning we only review data from all patients born between 2000 and 2018, who had been diagnosed with cPKU, mMPKU, or MHP, based on S-Phe values.  Among the 360,000 samples collected at the NBS, 72 HFA patients in need of a low-Phe diet were found. For 70 individuals, data on the Phe value in the initial DBS were available. Not all of these patients had their genotypes determined.

In this respect, it is important to calculate the incidence of PKU in this cohort and compare to other, e.g. European cohorts. To make a proper comparison, the authors must scrutinize these patients in the same way they did in table 1.

Reply: Incidence of all those on a diet is 1/5000. We are having difficulty comparing this to the others because this one is based on the true positives only, i.e. just those that required a diet. We published an article in 2013 (Grošelj, U.; Tanšek, M.Ž.; Podkrajšek, K.T.; Battelino, T. Genetske in klinične značilnosti bolnikov s fenilketonurijo v Sloveniji. Zdr. Vestn. 2013, 82, 767–777.), where calculated incidence of PKU (without HFA) in Slovenia is 1/6,000, cPKU 1/10,0000 and HFA 1/3,500).

  • Ln 78. “Accordingly, most 76 of the NBS programs globally are missing a substantial part of the individual MHP, be-77 cause it is considered that it is clinically not important to detect them. The main goal of 78 most NBS programs is thus not to detect all of the individuals with MHP, but not to miss 79 those who eventually need dietary intervention.” This seems to be exactly what the described program is doing with the low cut off.

Reply: We agree with your point. Because it was most likely written in an imprecise manner, we deleted it from the article. What we intended to express is - as we already argued above; this notion was among main reasons, why we conducted this study. We are ethically obliged to adjust the cut-off to the level so it not just detects all the patients eventually needing dietary therapy, but also that it burdens as few other families as possible. Based on our study we already adjusted Phe cut-off for our program, but we would also like to possibly help any other programs to possibly adjust their cut-off or at least to start thinking if their cut-off is really adequate. Based on the great spectrum of cut-offs globally, many of the programs probably have much room for improvement, because all of them cannot be right at the same time.

  • Ln 90. “Phe values have not yet reached their highest values [4].” But at a sampling time of 48-72 hrs they have.

Reply: Phe values in the earliest days or even weeks after birth are still raising and are highly dependent on the newborn's nutritional state. This statement was taken directly from the publication: Blau, N.; Van Spronsen, F.J.; Levy, H.L. Phenylketonuria. Lancet 2010, 376, 1417–1427, doi:10.1016/S0140-6736(10)60961-0.

“Classification is not always straightforward because phenylalanine concentrations are measured in newborn babies when blood phenylalanine might not have had time to reach its highest value.”

  • Ln 91: “in our study, the PKU classification was made based on APV when possible”. This is in contradiction with 2.3.3.

Reply: We concur with your remark; this statement was written in an incorrect manner, and was removed from the paper. There is likely some misunderstanding because we are discussing the PKU classification based on s-Phe and the true / false positive classification based on a diet. In the latter, we substituted the term classification for definition. We hope this clarifies things.

  • Ln 94: “We consider a false positive result when an additional sample is required, which is later found to be normal or screening-positive patients with Phe levels in areas where no dietary changes are needed. This is very important as it defines positives and influences the entire study – what are the Phe levels where no dietary changes are needed?

Reply: Usually, diets are initiated when s-Phe levels exceed 360 µmol/L as per international guidelines and available evidence (https://www.ncbi.nlm.nih.gov/pmc/articles/PMC5639803/).

  • Ln99-100: re testing after a high Phe result is not common in many programs. Most programs refer on the primary sample.

Reply: We concur; this sentence is written incorrectly; we changed it (please see page 9, line 95). “From the physician and laboratory analyst's point of view, follow-up for each such result requires contacting the clinic for additional samples, testing it, which requires additional cost, and then notifying the clinic of the second result.”

Ln 144-146. “The lower than the usual cut-144 off (120 μmol/L) in our program as compared to most”. You want to miss the cases with mild Phe elevations and being able to experiment with cut off values is not the reason the 120 was chosen and shouldn't be. The reason to do screening is not to do research, but to properly screen. The authors should indicate what the original cut off was based on. The point that they raise the cut off is valid, as it was low by international comparison. Their reasoning was off and still is.

Reply: The reason for the cut-off applied was not doing research, on the contrary; this research study was instrumental and basis for us to safely raise the cut-off! This was the main intention and not doing so would be according to our belief unethical. We hope that some other countries with non-optimal cut offs would follow us. As previously indicated, the initial cut-off was based on a pilot study using 7000 NBS samples and the kit manufacturer's recommendation. This value has been in use for almost thirty years and has never been revised (mostly due to complex organisational reasons, which were finally overcome at first successful expansion of the newborn screening program in last 3 years). The primary reasons for changing the cut-off are that the value of 120 µmol/L has remained unchanged for 30 years, that many other countries have higher cut-offs (where they almost certainly miss MHP, who do not require a diet anyway), and that we had an extremely large number of recalculations, particularly because the majority of them had a value of 120 µmol/L.

The value of 120 was not chosen for the purpose of this study but has been used for 30 years in the NBS in Slovenia. We intended to set up a new limit, with the primary objective of identifying everyone who needs treatment while also lowering the number of those who do not.

I think a complete re-writing and simplification of this paper is eminent. A proper line could be• We historically applied a cut-off value of 120 ug/l blood (based on....) • We had indications that this was not OK (mention indications) • We tried higher cut off in our historic data (one cohort). • Based on the results together we came up with a new cut-off of... Where you use a single definition of a positive case and describe all 72+16 = 88 patients in a table (or supplementary table).

Reply: Wherever feasible, we have used the proposed framing sentences.

). Unfortunately, the retrospective cohort could not be presented to the same level of detail, as we were able to present in prospective cohort, where we planned data collection in advance.